# Gene regulatory network plasticity predates a switch in function of a conserved transcription regulator

Isabel Nocedal[1,2*†], Eugenio Mancera[1,2‡], Alexander D Johnson[1,2*]

[1]Department of Microbiology and Immunology, University of California, San Francisco, United States; [2]Department of Biochemistry and Biophysics, University of California, San Francisco, United States

**Abstract** The rewiring of gene regulatory networks can generate phenotypic novelty. It remains an open question, however, how the large number of connections needed to form a novel network arise over evolutionary time. Here, we address this question using the network controlled by the fungal transcription regulator Ndt80. This conserved protein has undergone a dramatic switch in function—from an ancestral role regulating sporulation to a derived role regulating biofilm formation. This switch in function corresponded to a large-scale rewiring of the genes regulated by Ndt80. However, we demonstrate that the Ndt80-target gene connections were undergoing extensive rewiring prior to the switch in Ndt80's regulatory function. We propose that extensive drift in the Ndt80 regulon allowed for the exploration of alternative network structures without a loss of ancestral function, thereby facilitating the formation of a network with a new function.

*For correspondence: inocedal@mit.edu (IN); ajohnson@cgl.ucsf.edu (ADJ)

Present address: †Department of Biology, Massachusetts Institute of Technology, Cambridge, United States; ‡Departamento de Ingeniería Genética, CINVESTAV Unidad Irapuato, Irapuato, México

Competing interests: The authors declare that no competing interests exist.

## Introduction

The emergence of novel traits has long fascinated evolutionary biologists, with many intriguing examples observed across the tree of life (*Pigliucci, 2008*; *Rieppel, 2001*; *Shubin et al., 2009*; *Wagner and Lynch, 2010*). Although the rewiring of gene regulatory networks over evolutionary time is recognized as a key source of variation responsible for the modification of complex phenotypes (*Carroll, 2005*; *Davidson and Erwin, 2006*; *Li and Johnson, 2010*; *Wray, 2007*), we still lack an understanding of how new, large gene regulatory networks controlling novel phenotypes arise over evolutionary time. Typically, gene regulatory networks are composed of 'master' transcription regulators and many downstream 'target' genes, whose expression is controlled by these regulators in response to environmental or developmental signals. It has been proposed that new gene regulatory networks arise from a combination of de novo genes as well as conserved genes that have undergone changes in regulation (*Wagner and Lynch, 2010*). Yet, it remains unclear how a large number of genes (whether old or new) can be brought together to form a new network.

To address these questions, we examined the evolutionary history of Ndt80, a sequence-specific DNA-binding protein that is deeply conserved across a large group of fungal species encompassing approximately 300 million years of diversity. In most of these species, Ndt80 controls meiosis and sporulation, the coupled processes that form a portion of the fungal sexual cycle (*Chu and Herskowitz, 1998*; *Xu et al., 1995*). However, in the narrow lineage leading to the human fungal pathogen species *Candida albicans*, Ndt80 acquired a new role as a master regulator of the gene regulatory network that controls the formation of biofilms, multicellular communities of surface-associated cells (*Figure 1A*) (*Nobile et al., 2012*). This newly evolved trait enables *C. albicans* to persist on mucosal surfaces and on implanted medical devices (*Bonhomme and d'Enfert, 2013*; *Kojic and Darouiche, 2004*; *Nobile et al., 2012*) and is responsible for many of the disease-

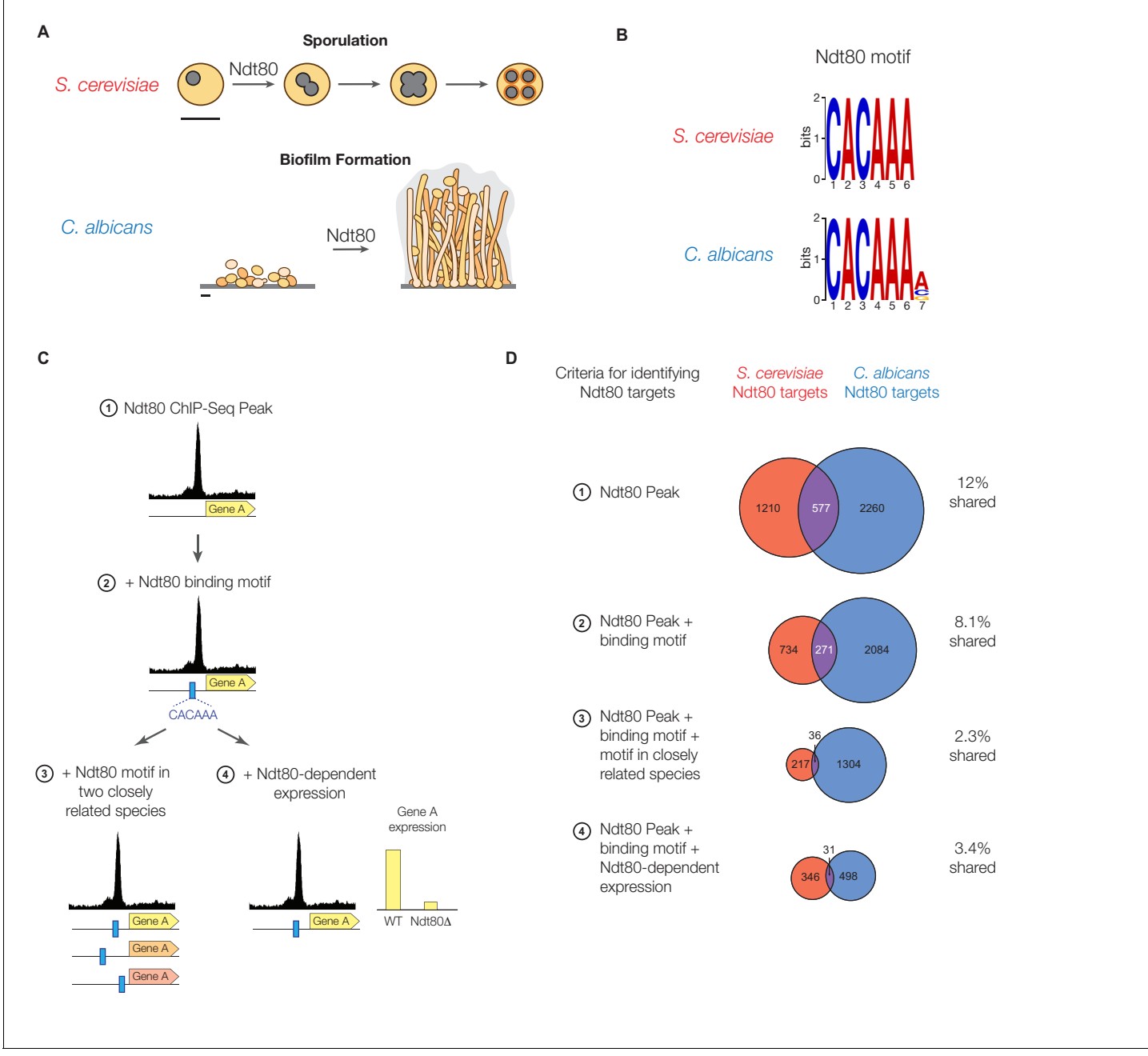

**Figure 1.** Ndt80 target genes differ between *S. cerevisiae* and *C. albicans*. (**A**) Diagram of sporulation in *S. cerevisiae* and biofilm formation in *C. albicans*. Scale bars represent 5 μm. (**B**) The *cis*-regulatory motif most highly enriched at locations of Ndt80 ChIP binding in *S. cerevisiae* and *C. albicans*. Motifs were generated independently for each species using DREME. The Ndt80 motifs determined de novo in this study closely match those identified previously for Ndt80 (*Chu and Herskowitz, 1998*; *Jolly et al., 2005*; *Nobile et al., 2012*). (**C**) Diagram of the four criteria used to identify Ndt80 regulatory targets. Criteria 1: significant ChIP-Seq enrichment in the intergenic region upstream of a gene relative to untagged control experiments. Criteria 2: ChIP-Seq enrichment and the presence of an Ndt80 motif in the intergenic region. Criteria 3: ChIP-Seq enrichment with the Ndt80 motif present in the intergenic region and also in orthologous intergenic regions of two very closely related species, suggesting the motif has been maintained by selection. Criteria 4: ChIP-Seq enrichment with the Ndt80 motif present in the intergenic region and Ndt80-dependent expression of the nearby gene, indicating that expression of the gene is under Ndt80 control. (**D**) Overlap in targets of *S. cerevisiae* Ndt80 (red) and *C. albicans* Ndt80B (blue), using the four different criteria from (**C**) to identify targets, when Ndt80 is highly expressed in each species (Materials and methods). Venn diagrams are roughly area-proportional (*Hulsen et al., 2008*).

The following figure supplements are available for figure 1:

**Figure supplement 1.** Ndt80 targets in *S. cerevisiae* and *C. albicans* when Ndt80 is endogenously expressed.

*Figure 1 continued on next page*

*Figure 1 continued*

**Figure supplement 2.** Ndt80 targets in *S. cerevisiae* and *C. albicans* considering only 1:1 orthologs between species.

**Figure supplement 3.** Ndt80 targets for paralogs Ndt80A and Ndt80B in *C. albicans*.

**Figure supplement 4.** Fraction of Ndt80 targets shared between biological replicates and between *S. cerevisiae* and *C. albicans*.

causing properties of *C. albicans* (*Nobile and Johnson, 2015*). The entire biofilm gene regulatory network in *C. albicans* is complex—Ndt80 alone controls the expression of hundreds of genes involved in biofilm formation—but it appears to have evolved relatively recently (*Nobile et al., 2012*). In this paper, we infer that approximately 20 to 100 million years ago (*Taylor and Berbee, 2006*), Ndt80's role changed from regulating sporulation and meiosis (its ancestral role) to regulating biofilm formation. We rigorously exclude a model in which the Ndt80 network remained relatively static until a sudden change occurred in the *Candida* clade when Ndt80's overall function changed from sporulation to biofilm formation. Instead, we found that the Ndt80 regulon was continuously undergoing significant rewiring in all lineages examined, even in those where the overall function of Ndt80 remained unchanged. Indeed, at the resolution of our experiments, the extent of Ndt80 rewiring was approximately the same whether its function had changed or not.

We propose that the inherent flexibility of the Ndt80 regulon facilitated the exploration of new regulatory networks, allowing it to reach positions in 'network space' that allowed for the evolution of a novel phenotype. This idea is analogous to examples of protein evolution in which mutational space can be sampled by drift without compromising the ancestral function of the protein. Such neutral excursions can then be exploited if they provide, as a by-product, a new function, for example weak catalysis of an alternative reaction by an enzyme (*Aharoni et al., 2005*; *Bridgham et al., 2006*; *Coyle et al., 2013*; *Khersonsky and Tawfik, 2010*; *Wagner, 2005a, 2005b*). Our work provides an empirical, case study of an analogous exploitation of regulatory network plasticity in the evolution of novel transcription regulator function.

It has been well established that transcription regulator-target gene interactions change over evolutionary timescales. For example, the regulators controlling the ribosomal genes and the **a**-specific genes in fungi have changed over several hundred million years of evolution (*Baker et al., 2012*; *Ihmels et al., 2005*; *Lavoie et al., 2010*; *Tanay et al., 2005*; *Tsong et al., 2006*) (*Sorrells et al., 2015*). Likewise, transcription regulators (such as Mcm1 and Ste12) have been shown to undergo extensive rewiring of their regulatory targets over similar timescales (*Borneman et al., 2007*; *Tuch et al., 2008*). Such rewiring is not restricted to fungi; for example, enhancers in *Drosophila* have been shown to undergo rapid changes in the nature and arrangement of their *cis*-regulatory sequences (*Frankel et al., 2012*; *Gompel et al., 2005*; *Ludwig et al., 2000*; *Swanson et al., 2011*). This high degree of network plasticity seems to be a general feature of many regulatory networks. Based on the case study of Ndt80, we propose that this plasticity can be captured to produce new, complex regulatory networks.

## Results

### Ndt80 target genes differ markedly between *S. cerevisiae* and *C. albicans*

As described above, the transcription factor Ndt80 is required for sporulation and meiosis in *S. cerevisiae* and biofilm formation in *C. albicans* (*Figure 1A*). In principle, the difference in phenotype could be due to a difference in the identity of the genes regulated by Ndt80 between these species; alternatively, Ndt80 could regulate the same gene set in both species and the difference in phenotype could be due to something else, for example, Ndt80 target genes acquiring new enzymatic functions or Ndt80 itself being regulated differently in the two species.

To distinguish between these models, we identified the genes directly regulated by Ndt80 in both *S. cerevisiae* and *C. albicans* using chromatin immunoprecipitation of epitope-tagged Ndt80

followed by high-throughput sequencing (ChIP-Seq). To minimize the effect of species-specific differences in the levels of Ndt80 expression (and to capture as much of the network as possible), the promoters of Ndt80 in both species were replaced with high-expression constitutive promoters (Materials and methods). It has been well established that, even using proper controls, chromatin immunoprecipitation produces false positives in addition to valid instances of binding (*Fan and Struhl, 2009*; *Teytelman et al., 2013*). To eliminate spurious signals and identify bona fide instances of Ndt80 binding, we employed four different criteria of increasing stringency (*Figure 1C*). First, we simply identified genes with significant peaks of ChIP enrichment in their intergenic regions; this is our least stringent criteria. Second, we filtered this set to include only those ChIP enrichment peaks that contained an Ndt80 *cis*-regulatory motif in the intergenic region (this motif is discussed in more detail below). Third, of those ChIP peaks that contained an Ndt80 motif, we further refined the set of Ndt80 targets by requiring that the motif also be present in the orthologous intergenic region of two very closely related species, as this greatly increases the likelihood that the Ndt80-binding site was maintained by selection. (For *S. cerevisiae*, we used *S. mikatae* and *S. kudriavzevii*, for *C. albicans* we used *C. tropicalis* and *C. dubliniensis* [*Byrne and Wolfe, 2005*; *Lohse et al., 2013*]). Lastly, we used gene expression data to restrict Ndt80 targets to ChIP peaks with an Ndt80 motif where the gene also exhibited a significant change in mRNA expression when Ndt80 was deleted (*Chu et al., 1998*; *Nobile et al., 2012*).

Each of the four criteria were used to identify and compare the genes regulated by Ndt80 between *S. cerevisiae* and *C. albicans*. While the number of genes identified as Ndt80 targets differs depending on the criteria used, all four methods produce the same overall conclusion: Ndt80 regulates many genes in each species, with very little overlap between them (*Figure 1D*). By all methods, fewer than 13% of the targets of Ndt80 in these two species are shared; with the most stringent criteria this drops to 3.4% (*Figure 1D*). To ensure that constitutive overexpression of Ndt80 was not responsible for this large difference in Ndt80 targets, we also performed ChIP-Seq on Ndt80 in both species under the control of the endogenous promoter. While many fewer regions are bound than with constitutive Ndt80 expression, we similarly find very little overlap in the Ndt80 targets in *S. cerevisiae* and *C. albicans* (less than 12%, *Figure 1—figure supplement 1*). We note that experimental noise is not sufficient to explain the difference in targets, as the differences we observe between biological replicates accounts for only a small fraction of the difference observed between *S. cerevisiae* and *C. albicans* (*Figure 1—figure supplement 4*).

These results all lead to the conclusion that Ndt80 regulates distinct sets of genes in *S. cerevisiae* and *C. albicans*. Most of these genes have 1:1 orthologs in both species, although there are also a smaller number of species-specific genes. If we exclude the species-specific genes from our analysis, we still find relatively little overlap in Ndt80 targets (4–21% targets shared, depending on target identification criteria used, *Figure 1—figure supplement 2*), showing that gene gains and losses alone cannot account for the change in Ndt80 targets between *S. cerevisiae* and *C. albicans*. Despite the significant differences in Ndt80 targets, however, the *cis*-regulatory sequence bound by Ndt80 in both species is highly conserved ([*Jolly et al., 2005*; *Nobile et al., 2012*], *Figure 1B*). This observation indicates that the Ndt80 regulon has been significantly rewired between *S. cerevisiae* and *C. albicans* without a change in the DNA-binding specificity of Ndt80.

## Divergence in Ndt80 targets not a result of Ndt80 gene duplication

While the *NDT80* genes in *S. cerevisiae* and *C. albicans* discussed above are homologous, *C. albicans* has an additional paralog of Ndt80, resulting from a gene duplication (*Figure 2A*, [*Sellam et al., 2010*]). The Ndt80 homolog whose targets we identified above will be referred to as Ndt80B, while the additional paralog will be referred to as Ndt80A (*Figure 2—figure supplement 1*). While Ndt80B has been shown to be required for biofilm formation in *C. albicans* (*Nobile et al., 2012*), the regulatory function of Ndt80A is unknown (deletion of Ndt80A has no effect on biofilm formation (Figure 4H, [*Nobile et al., 2012*]) or any other phenotype tested). To test the possibility that the difference in targets between Ndt80 in *S. cerevisiae* and Ndt80B in *C. albicans* is simply a consequence of this gene duplication, we identified the regulatory targets of Ndt80A in *C. albicans* by ChIP-Seq under control of the same constitutive promoter used for Ndt80B. We found that the targets regulated by Ndt80A represent a small subset of the targets of Ndt80B (9–11%, *Figure 1—figure supplement 3*); that is, Ndt80A binds to many fewer genomic regions, but all these regions are also bound by Ndt80B. If we compare the targets of Ndt80 in *S. cerevisiae* to the targets of

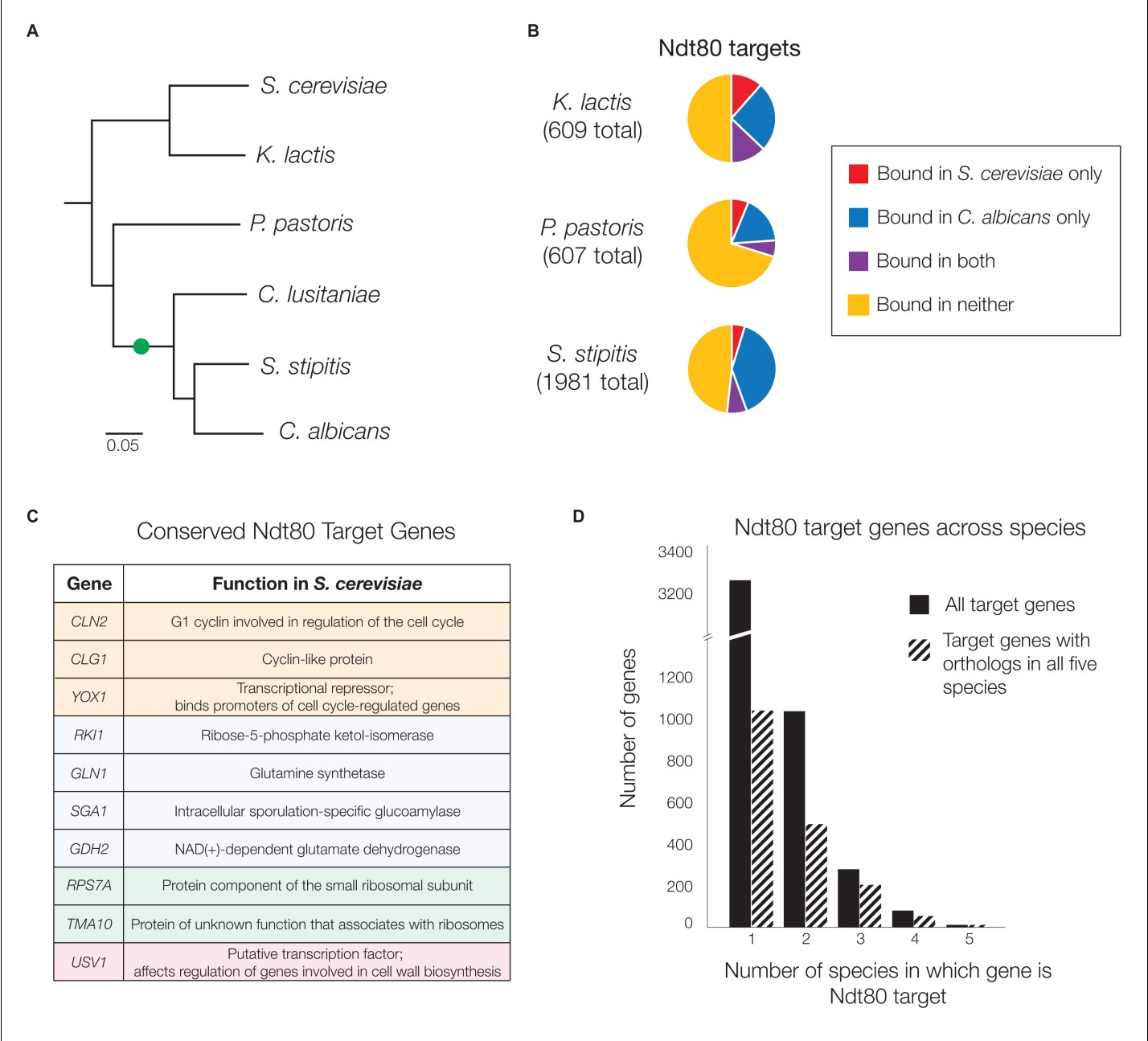

**Figure 2.** Ndt80 target genes differ across species descending from the *S. cerevisiae* - *C. albicans* common ancestor. (**A**) Phylogenetic tree of the species investigated, inferred from protein sequences of 73 highly conserved genes (***Lohse et al., 2013***) with the scale representing the number of substitutions per site. The likely position of the *NDT80* gene duplication is indicated with a green circle. Ndt80 protein tree shown in ***Figure 2—figure supplement 1***. (**B–D**) Ndt80 targets identified by Criteria 2 (***Figure 1C***). (**B**) The proportion of Ndt80 targets in *K. lactis*, *P. pastoris*, and *S. stipitis* that are shared with the Ndt80 targets in *S. cerevisiae*, *C. albicans*, or both. The proportion of targets shared, using all four different criteria for identifying targets (***Figure 1C***) shown in ***Figure 2—figure supplement 2***. (**C**) List of genes bound by Ndt80 in all five species tested with functional annotations (color-coded) from *S. cerevisiae,* as described in the text (SGD, [***Cherry et al., 2012***]). (**D**) Histogram of all Ndt80 targets in the five species tested (*S. cerevisiae, K. lactis, P. pastoris, S. stipitis,* and *C. albicans*) according to the number of species in which that gene is a target. All target genes (black) and those with 1:1 orthologs across all five species (dashed) are shown. For (**B–D**), the union of Ndt80A and Ndt80B bound genes were used for *S. stipitis* and *C. albicans*. The *cis*-regulatory site bound by Ndt80 in each species shown in ***Figure 2—figure supplement 4***.

The following figure supplements are available for figure 2:

**Figure supplement 1.** Similarity of Ndt80 protein sequences across many yeast species.

*Figure 2 continued on next page*

*Figure 2 continued*

**Figure supplement 2.** Ndt80 targets in *K. lactis*, *P. pastoris*, and *S. stipitis* categorized according to overlap with *S. cerevisiae* and *C. albicans*, using all four methods of target identification.

**Figure supplement 3.** Histogram of simulated overlap in Ndt80 targets across five species.

**Figure supplement 4.** Ndt80 *cis*-regulatory motifs identified in ChIP-Seq experiments.

Ndt80A in *C. albicans*, we find that less than 4% of the overall targets are shared, regardless of target identification criteria (*Figure 1—figure supplement 3*), demonstrating that Ndt80A's targets are no more similar to the targets of Ndt80 in *S. cerevisiae* than that of Ndt80B. These experiments demonstrate that both of the Ndt80 paralogs in *C. albicans* have undergone significant rewiring since *S. cerevisiae* and *C. albicans* diverged, indicating that the Ndt80 gene duplication was not directly responsible for the divergence in targets between these two species.

## Ndt80 target genes also differ between *S. cerevisiae*, *K. lactis*, *P. pastoris*, *S. stipitis*, and *C. albicans*

To reconstruct a timeline of the evolution of the Ndt80 gene regulatory network, we conducted ChIP-Seq experiments in three additional species that branch from the common ancestor of *S. cerevisiae* and *C. albicans* at highly informative points: *Kluyveromyces lactis*, *Pichia pastoris,* and *Scheffersomyces stipitis* (*Figure 2A*). In each case, we used the strategy of expressing Ndt80 under the control of a strong constitutive promoter to minimize expression differences between species. In *C. albicans* and *S. stipitis,* targets were identified for the two Ndt80 paralogs separately, and found to be highly overlapping (*Figure 1—figure supplement 3*, *Supplementary file 1*). Thus, the union of Ndt80 paralog targets were taken to represent all Ndt80 targets in those species.

We first investigated whether the regulatory targets of Ndt80 in each of these species more closely resemble the targets in *S. cerevisiae* or the targets in *C. albicans*. Ndt80 targets were identified in all species based on the criteria of ChIP enrichment plus Ndt80 motif presence (*Figure 1C*), and compared to the targets of Ndt80 in *S. cerevisiae* and *C. albicans*. Counter to our initial expectation, we found that a very large number of Ndt80 targets in each species are not targets of Ndt80 in either *S. cerevisiae* or *C. albicans* (50% of all targets in *K. lactis*, 70% in *P. pastoris*, and 48% in *S. stipitis*, *Figure 2B*). Thus, the Ndt80 regulons in these three species do not closely resemble those of either *S. cerevisiae* or *C. albicans*; instead, each have acquired a distinctive set of Ndt80 target genes. If we repeat this analysis using alternative criteria for identifying Ndt80 targets (*Figure 1C*), this conclusion still holds (*Figure 2—figure supplement 2*), indicating that it is robust to the stringency of Ndt80 target identification.

Consistent with the idea that Ndt80 has acquired a unique set of targets in each species, only ten genes are targets of Ndt80 in all five species tested (*Figure 2C*). While this is more than expected strictly by chance considering the number of regions bound in each species (p<10$^{-5}$, *Figure 2—figure supplement 3*), this number pales in comparison to the 3261 genes that are Ndt80 targets in just one of the five species (*Figure 2D*). Even if we consider only genes with 1:1 orthologs across all five species, there are more targets bound in only one species (1041) than bound in two or more species (772) (*Figure 2D*). In short, the Ndt80 regulon differs extensively between each of the five species tested.

## Ndt80-binding differences are largely determined by the gain and loss of *cis*-regulatory sites

Because the Ndt80-target gene connections differ significantly between species, we considered whether these differences were due primarily to changes in the Ndt80 protein itself, or to changes in the distribution of its *cis*-regulatory motif across the different genomes. Although the *cis*-regulatory sequence recognized by Ndt80 is conserved across these species (*Figure 2—figure supplement 4*), changes in the Ndt80 protein itself that alter, for example, protein-protein interactions with other transcription regulators, could account for many of the differences between species. The sequence

of the Ndt80 protein does differ considerably across the species tested, with only 55% similarity in the DNA-binding domain between *S. cerevisiae* and *C. albicans* (*Sellam et al., 2010*), and 35% similarity in the proteins overall (*Figure 2—figure supplement 1*).

To test whether these changes in protein sequence were primarily responsible for the observed differences in Ndt80 target genes between species, we expressed the *P. pastoris NDT80* gene in *S. cerevisiae* under the control of the same constitutive promoter used for ChIP-Seq of the endogenous *S. cerevisiae NDT80*, and carried out ChIP-Seq on the *P. pastoris NDT80* (*Figure 3A*). If we compare Ndt80 ChIP enrichment genome-wide in this experiment to Ndt80 enrichment for the native Ndt80 in each species, we find a much stronger correlation between genomic binding of the heterologous *P. pastoris* Ndt80 and the native *S. cerevisiae* Ndt80 (*Figure 3B*) than between the heterologous *P. pastoris* Ndt80 and the native *P. pastoris* Ndt80 (*Figure 3C*). Similarly, we find more shared target genes between the different Ndt80 proteins expressed in *S. cerevisiae* (75 genes, *Figure 3—figure supplement 1*) than between the same *P. pastoris* Ndt80 protein expressed in different species (4 genes, *Figure 3—figure supplement 1*). These results demonstrate that changes in the Ndt80 protein sequence across species have had only a minor effect on the Ndt80-target gene connections; instead, the connections are predominantly determined by the distribution of *cis*-regulatory sequences across the different genomes.

Although it seems likely that gains and losses of Ndt80 *cis*-regulatory sequences are largely responsible, we cannot formally exclude the contributions of changes in *cis*-regulatory sequences for other (thus far, unidentified) proteins that might help recruit Ndt80 to DNA. These observations on the key role of *cis*-regulatory change do not mean that changes in the protein are unimportant; indeed, the heterologous *P. pastoris* Ndt80 does not complement the sporulation defect observed in an *ndt80* deletion in *S. cerevisiae* (data not shown), nor does it occupy all the genome positions

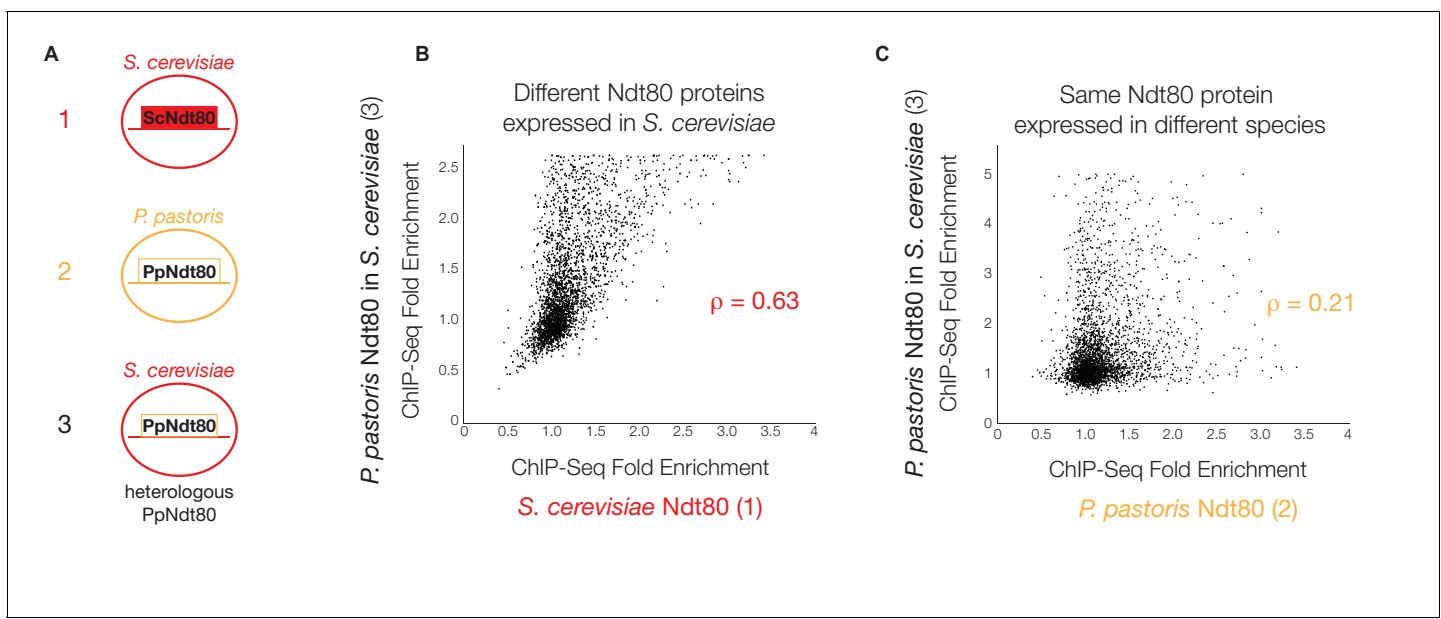

**Figure 3.** *P. pastoris* Ndt80 binds to regions bound by *S. cerevisiae* Ndt80 when expressed in *S. cerevisiae*. (**A**) Diagram of strains used to generate data in (**B**) and (**C**): (1) tagged native Ndt80 in *S. cerevisiae*, (2) tagged native Ndt80 in *P. pastoris*, and (3) tagged heterologous *P. pastoris* Ndt80 expressed in *S. cerevisiae*. (**B and C**) Comparisons of the maximum ChIP-Seq fold enrichment for the intergenic regions of all genes with 1:1 orthologs in *S. cerevisiae* and *P. pastoris*, with Spearman's rank correlation coefficient shown (p-value=$7.2 \times 10^{-117}$ for (**B**), $1.02 \times 10^{-296}$ for (**C**)). Numbers in axis labels correspond to simplified diagrams in (**A**). (**B**) *S. cerevisiae* Ndt80 vs. *P. pastoris* Ndt80 expressed in *S. cerevisiae*. (**C**) *P. pastoris* Ndt80 expressed in *P. pastoris* vs. *P. pastoris* Ndt80 expressed in *S. cerevisiae*. A comparison of Ndt80 targets identified in each experiment using Criteria 1 shown in *Figure 3—figure supplement 1*.

The following figure supplement is available for figure 3:

**Figure supplement 1.** Heterologous *P. pastoris* Ndt80 targets compared to native *P. pastoris* and *S. cerevisiae* Ndt80 targets.

characteristic of the *S. cerevisiae* protein. Our results do show, however, that the heterologous *P. pastoris* protein does occupy a substantial fraction of the *S. cerevisiae*-specific positions along the genome. This observation strongly supports the conclusion that the majority of the differences in the Ndt80 regulons across species are due to gains and losses of *cis*-regulatory sequences.

## Ndt80 is required for sporulation in *K. lactis*, *P. pastoris*, and *C. lusitaniae*

To determine whether the rewiring of the Ndt80 gene regulatory network corresponds to the changes in the overall function of Ndt80, we determined the phenotype of an *ndt80* deletion mutant in three species that branch between *S. cerevisiae* and *C. albicans*: *K. lactis*, *P. pastoris,* and *C. lusitaniae* (*Figure 2A*). We first tested whether this deletion had an effect on sporulation and found, in *K. lactis* and *P. pastoris,* that the *ndt80* deletion mutant is completely deficient in its ability to sporulate (*Figure 4A and B*). *C. lusitaniae*, like *C. albicans*, has two paralogs of Ndt80. Deletion of one paralog (Ndt80A) results in a complete deficiency in sporulation, while deletion of the other paralog (Ndt80B) results in a significant reduction in sporulation efficiency (*Figure 4C*). In contrast to these four species, *C. albicans* has never been observed to undergo sporulation and meiosis, relying instead on an alternate parasexual cycle (*Bennett and Johnson, 2003*; *Butler et al., 2009*). Given that Ndt80 is required for sporulation in *S. cerevisiae* (*Chu and Herskowitz, 1998*; *Xu et al., 1995*), *K. lactis*, *P. pastoris*, and *C. lusitaniae*, and given the phylogenetic relationship between these species (*Figure 2A*), the most parsimonious model is that, in the ancestor of these species, Ndt80 regulated sporulation.

We next investigated the requirement for Ndt80 in biofilm formation in these species. In *C. albicans*, biofilms consist of thick structures of different cell types (yeast-form and hyphae) that can form both in vitro and in vivo [*Andes et al., 2004*]). Deletion of *NDT80B* in *C. albicans* causes a severe defect in surface adherence and biofilm formation (*Figure 4H*). In contrast, in *S. cerevisiae*, *K. lactis*, *P. pastoris*, and *C. lusitaniae*, the *ndt80* mutants and wild-type are comparable in their ability to adhere to a solid surface (*Figure 4D–G*). It is worth noting that *S. cerevisiae*, *K. lactis* and *P. pastoris* form only a thin layer of yeast-form cells on the solid surface rather than the type of thick, multicellular biofilms characteristic of those formed in *C. albicans,* while *C. lusitaniae* forms a biofilm somewhat thinner than that of *C. albicans*. These results highlight the specialized type of biofilm produced by *C. albicans* and the pivotal role played by Ndt80 in its evolution.

Taken together, these results indicate that Ndt80 regulated sporulation in the shared ancestor of *S. cerevisiae* and *C. albicans*, and that it gained a role in biofilm formation along the *C. albicans* lineage, after it branched off from *C. lusitaniae* (*Figure 2A*). We believe this scenario is more likely than that of the next most parsimonious model, which holds that Ndt80 regulated both sporulation and biofilm formation in the shared ancestor, as this would require at least three independent losses of biofilm regulation. Given that Ndt80 regulates hundreds of genes in each species, a single gain of Ndt80 function seem more plausible than three independent losses.

Combining this phenotypic data with the Ndt80 target identification previously discussed, we can begin to pinpoint the importance of individual genes in the overall switch in Ndt80 function. Only 10 genes are targets of Ndt80 in all five species tested (*S. cerevisiae*, *K. lactis*, *P. pastoris*, *S. stipitis*, and *C. albicans*) (*Figure 2C*). This suggests that a very limited set of genes may have been repurposed from sporulation to biofilm formation. Notably, five of these genes (*CLN2*, *CLG1*, *GLN1*, *GDH2*, *RPS7a*) are strongly upregulated in biofilm formation (*Fox et al., 2015*). If we examine the 10 shared genes based on their known functions in *S. cerevisiae*, three are involved in cell cycle regulation (*CLG1*, *CLN2*, and *YOX1*), four are metabolic enzymes (*GDH2*, *GLN1*, *RKI1*, and *SGA1*), two are proteins associated with the ribosome (*RPS7a* and *TMA10*), and one is involved in regulation of cell wall biosynthesis (*USV1*) (*Figure 2C*). The protein Usv1 is particularly intriguing, as its ortholog in *C. albicans*, Bcr1, is one of the other master regulators of biofilm formation (*Nobile et al., 2012*; *Nobile and Mitchell, 2005*). This observation suggests that two of the master regulators of biofilm formation, Ndt80 and Bcr1, may have had an ancestral regulatory relationship that was co-opted in the evolution of the biofilm regulatory network. In summary, while there are small vestiges of the sporulation network present in the *C. albicans* biofilm network, the great majority of connections are new.

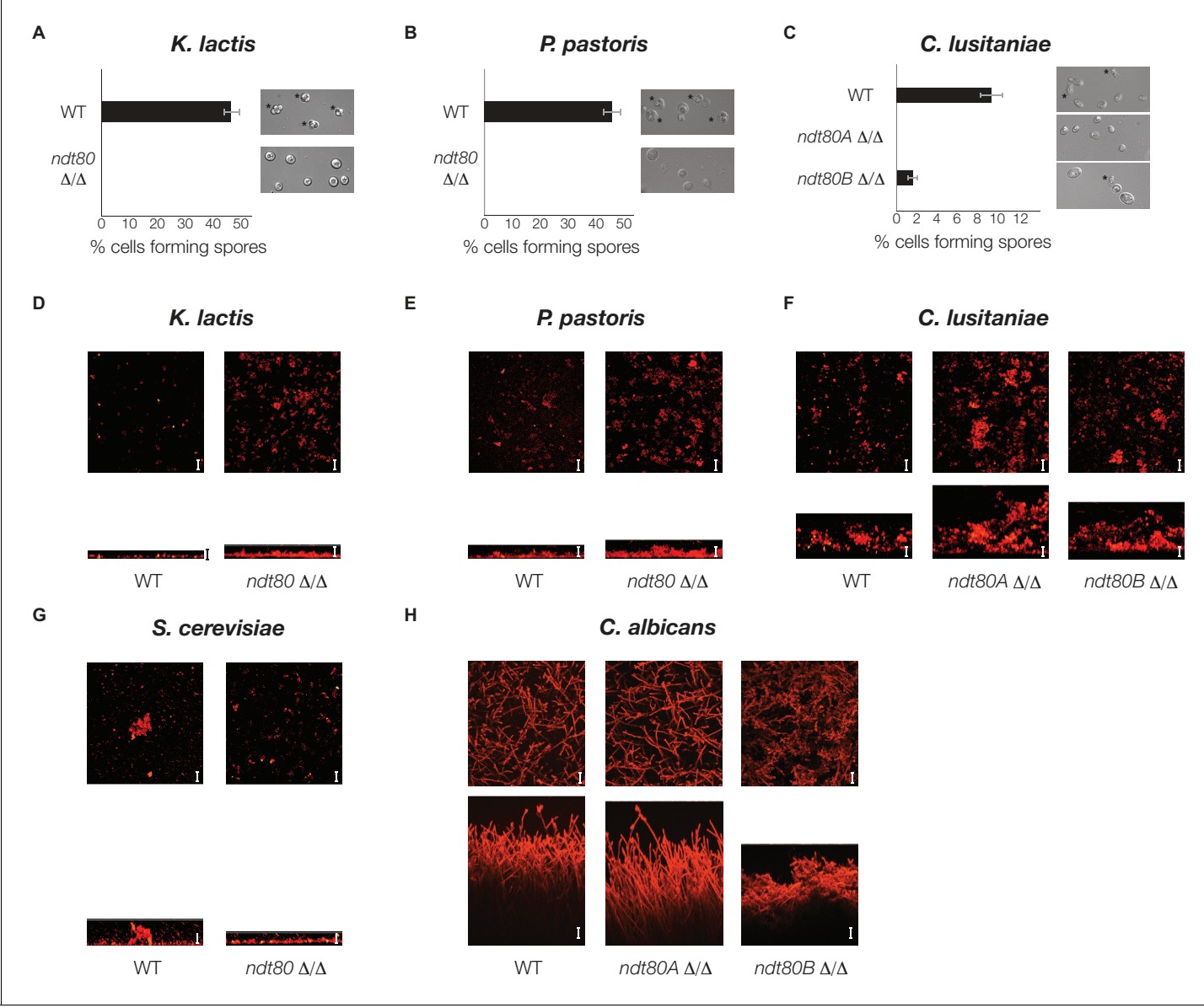

**Figure 4.** Ndt80 is required for sporulation but is dispensable for biofilm formation, in *K. lactis*, *P. pastoris*, and *C. lusitaniae*. (A–C) Light microscope images of genetically matched wild-type and *ndt80* deletion strains (Stars indicate diploid cells that have undergone sporulation) and quantification of the percent of cells exhibiting spores, as measured by microscopy (200 cells counted for each strain). (D–H) Confocal scanning laser microscopy images of biofilm formation for genetically matched wild-type and *ndt80* deletion strains. Top view of biofilm shown above side view for each, with scale bars representing 25 μm.

## The transcription network that controls meiosis and sporulation changes extensively

Because the targets of Ndt80 appear to have rewired significantly during the switch from sporulation regulation to biofilm regulation, we tested whether Ndt80 targets are more similar among species with a conserved Ndt80 phenotype than among species with a diverged Ndt80 phenotype. If so, this would suggest that the dramatic rewiring of targets was associated with the switch in Ndt80 regulatory function. We focused on the four species for which we know both the Ndt80 phenotype and the binding targets: *S. cerevisiae*, *K. lactis*, *P. pastoris*, and *C. albicans*. For every two-species pair, we compared the Ndt80 targets, considering only genes with 1:1 orthologs for both species. We find relatively little overlap in Ndt80 targets for any of these comparisons (9–14%), but more

importantly, we do not find a correlation between target overlap and conservation of overall Ndt80 function (*Figure 5A*). For example, a larger fraction of targets (14%) are shared between *S. cerevisiae* and *C. albicans* than between *S. cerevisiae* and *P. pastoris* (9.2%). If we take into account the different divergence times for each of these two-species comparisons (normalizing to the divergence time between *S. cerevisiae* and *K. lactis*), we still find that conservation of overall Ndt80 function does not correlate with increased conservation of Ndt80 targets (*Figure 5B*). This pattern also holds if we use more stringent criteria to identify Ndt80 targets in each species (*Figure 5—figure supplement 1*). Overall, these results show that the targets of Ndt80 change extensively, even while Ndt80's conserved role in regulating sporulation is maintained (*Figure 5C*). While we cannot entirely

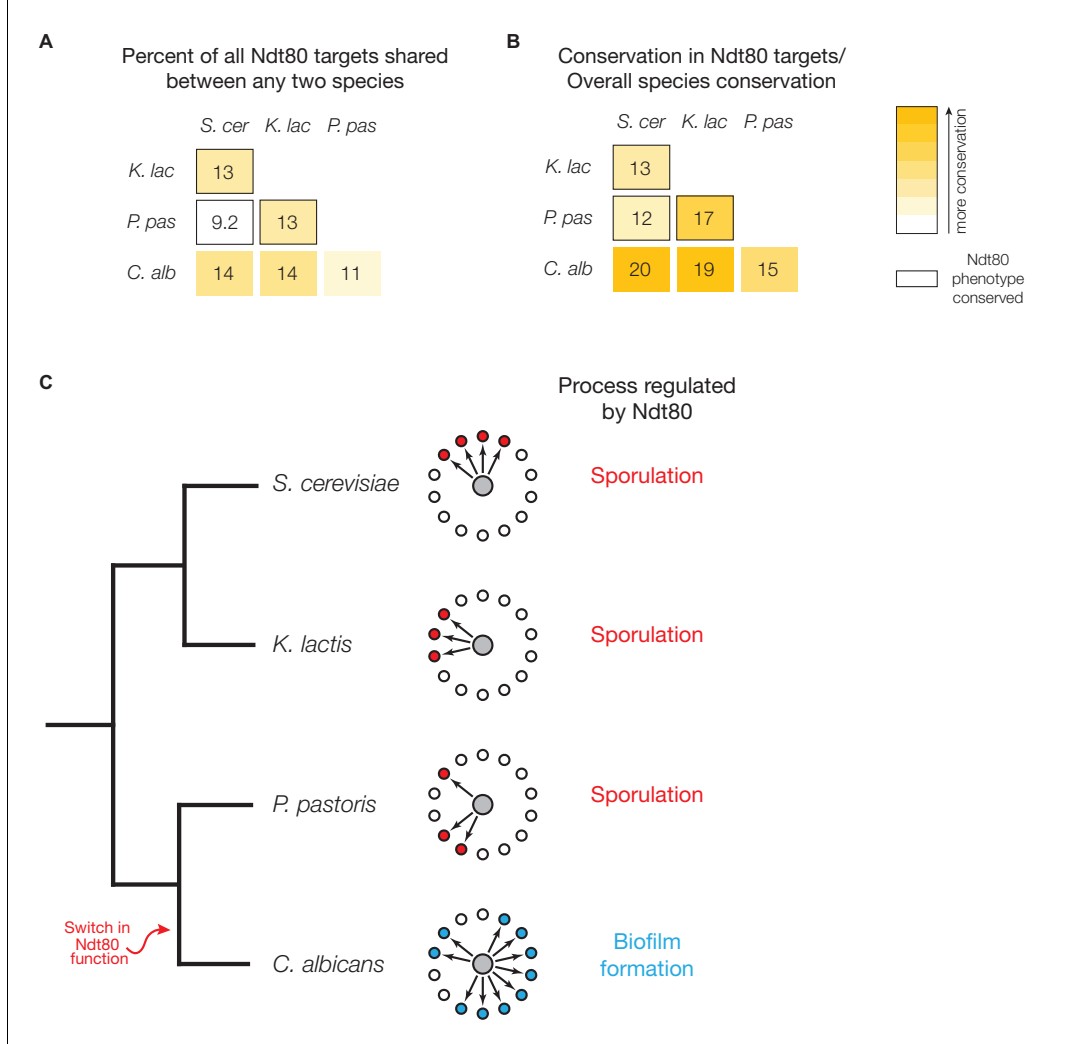

**Figure 5.** The targets of Ndt80 differ among species even with a conserved Ndt80 phenotype. (**A**) Percent of all Ndt80-bound genes (identified using Criteria 2) shared between any two species tested; for this analysis, only 1:1 orthologs in each two-species comparison were considered. (**B**) Percent of all Ndt80-bound genes shared between any two species, normalized to the genome-wide substitutions per site (*S. cerevisiae – K. lactis* set to 1). Species comparisons between two species with a conserved Ndt80 phenotype shown with black outline for (**A**) and (**B**). Percent overlap and normalized percent overlap for targets identified using all four criteria (*Figure 1C*) shown in *Figure 5—figure supplement 1*. (**C**) Cladogram of species for which Ndt80 phenotype and target genes have been identified, with simplified cartoons representing Ndt80 ChIP-Seq targets in each species. The grey circle in the center represents Ndt80, while the smaller circles represent genes present in all four species, with each small circle representing ~100 genes. The arrows indicate that Ndt80 binds to those genes in that species.

The following figure supplement is available for figure 5:

**Figure supplement 1.** Percent overlap in Ndt80 targets across different species, using all four methods of target identification.

rule out the possibility that Ndt80 has another, unknown regulatory function in one or more of these species that accounts for the significant difference in Ndt80 targets, we know that *S. cerevisiae* Ndt80 is produced specifically during sporulation (*Chu and Herskowitz, 1998*). Similarly, by performing transcription profiling, we find that in *K. lactis* and *P. pastoris*, Ndt80 is not expressed in mitotic cells (*Supplementary file 1*), consistent with the idea that Ndt80 is primarily needed during sporulation and is unlikely to have additional regulatory roles.

The rewiring of Ndt80 targets across species in which Ndt80 is known to regulate sporulation indicates that only a small number of Ndt80 targets must be maintained in order for the sporulation network to be functional. Although there are several possible explanations for how Ndt80 could have maintained a role in sporulation despite the near complete turnover of its target genes, we tested perhaps the most intriguing hypothesis: the genes required for sporulation and meiosis are themselves changing across species. To test this hypothesis, we measured global gene expression during sporulation and mitotic growth in *K. lactis* and *P. pastoris* (Materials and methods). We compared the genes upregulated during sporulation in these species to those upregulated during sporulation in *S. cerevisiae* (*Chu et al., 1998*) and found extensive differences in the genes activated during sporulation across these species (*Figure 6A*). Only 25 genes show sporulation-activation across *S. cerevisiae*, *K. lactis*, and *P. pastoris*, while 577 genes are activated uniquely in only one of the three species. Of the 65 genes known to be required for sporulation in *S. cerevisiae* that also exhibit sporulation-specific activation, only nine are activated during sporulation in both *K. lactis* and *P. pastoris*, with 21 of these genes missing altogether from the genomes of one or both of these species (*Figure 6—figure supplement 2*). We also measured global gene expression in an *ndt80* deletion mutant in both *K. lactis* and *P. pastoris* under sporulation conditions, and found that the genes whose expression depends on Ndt80 during sporulation also differ significantly across species, with only seven genes showing Ndt80-dependent expression in all three species (*Figure 6B*, *Figure 6—figure supplement 1*). We conclude that there are significant differences in sporulation-specific gene expression among *S. cerevisiae*, *K. lactis*, and *P. pastoris*, and that this contributes significantly to the large differences observed in Ndt80 targets across species.

## Discussion

Using experiments performed in six different extant yeast species, we have deduced that the transcription regulator Ndt80 underwent extensive rewiring of its regulatory connections before it became incorporated into a newly evolving regulatory network. This incorporation, which occurred on the lineage leading to *C. albicans*, resulted in an overall switch in Ndt80's function from an ancestral role in regulating sporulation and meiosis to a derived role in regulating biofilm formation. We have shown that this switch in function was not accompanied by a change in the DNA-binding specificity of Ndt80 (*Figure 1B*, *Figure 2—figure supplement 4*); rather, a change in the distribution of the conserved Ndt80 *cis*-regulatory sequence across hundreds of genes in the genome largely accounts for the different functions of Ndt80 in *S. cerevisiae* and *C. albicans*.

A priori, we considered two likely explanations for the change in Ndt80 function. First, we hypothesized that a duplication of Ndt80 in the *C. albicans* clade could have allowed the ancestral function (sporulation) to be conserved in one paralog, while the other acquired a new function (biofilm regulation). However, the similarity in function of Ndt80 paralogs in *C. lusitaniae* (*Figure 4C and F*) as well as the similarity in targets of Ndt80 paralogs in *C. albicans* (*Figure 1—figure supplement 3*) rule out this model. A second plausible model held that a sudden change in Ndt80-target gene connections occurred along the *C. albicans* lineage, perhaps triggered by the loss of meiosis in this clade, relaxing constraints on Ndt80 target gene connections and allowing old connections to rapidly break and new connections to rapidly form. However, we also ruled out this model by showing that similarly high rates of Ndt80 rewiring occurred in all phylogenetic branches examined, even those where the function of Ndt80 remains conserved in sporulation (*Figure 5A and B*). Rather than a sudden shift in regulation, our results indicate continuous flexibility in the regulon of Ndt80, allowing it to sample many configurations even while retaining its ancestral function. In support of this idea, we showed that genes upregulated during sporulation (by Ndt80 and other regulators of sporulation) vary considerably across different species (*Figure 6*), indicating that many different network configurations can support sporulation. Consistent with this idea, it has previously been noted that several key genes required for meiosis and sporulation in *S. cerevisiae* (*IME1*, *ZIP2*, *SPO13*) are

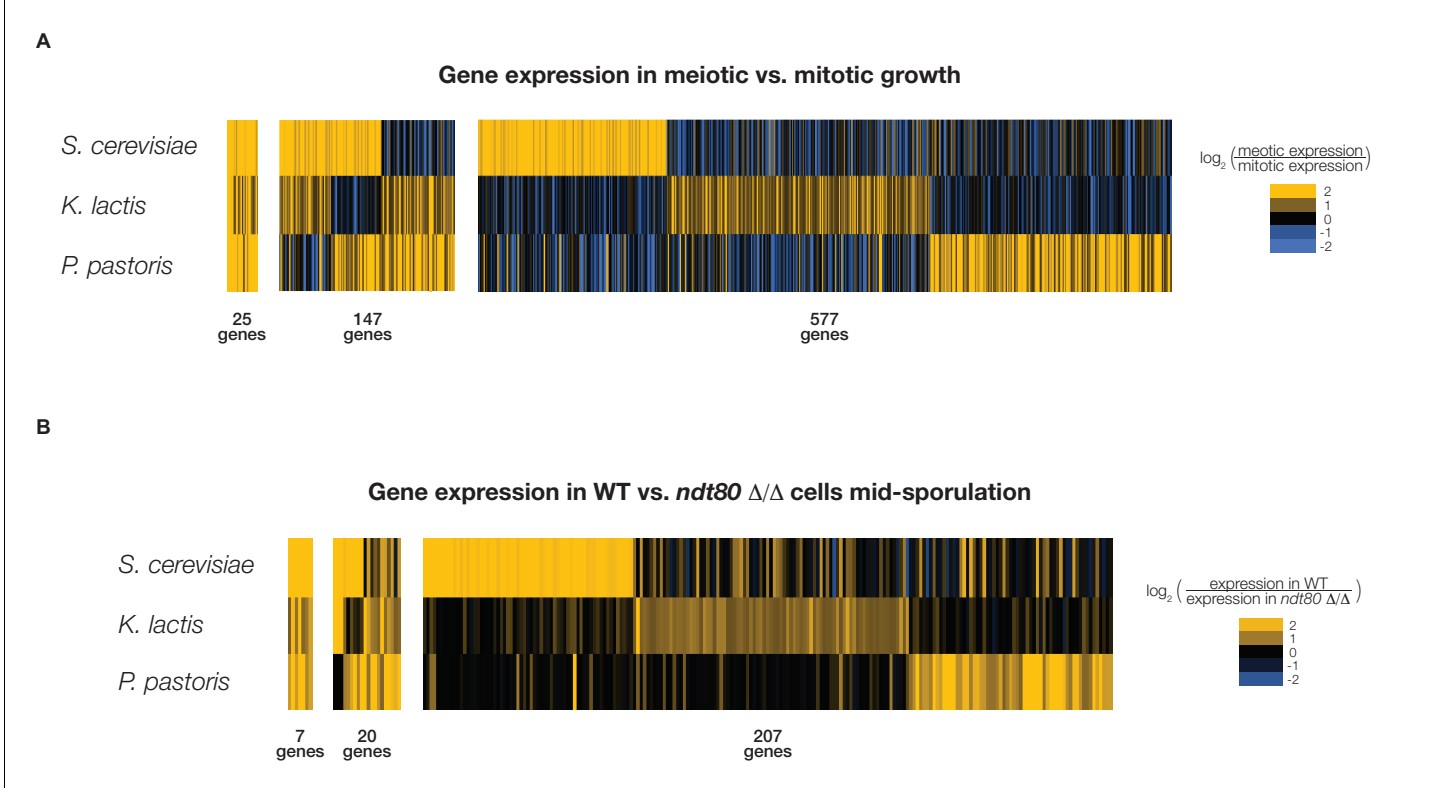

**Figure 6.** Genes induced during sporulation differ significantly across *S. cerevisiae*, *K. lactis*, and *P. pastoris*. (**A**) Gene expression in meiotic growth compared to mitotic growth. Genes with significant upregulation in meiosis in at least one species shown. (**B**) Gene expression in a wild-type strain compared with an *ndt80* △/△ strain for all three species. Genes with significant upregulation in wild-type in at least one species shown. For (**A**) and (**B**), each line represents a single gene with the color of the line representing the ratio of expression in the two conditions. Numbers below represent the number of genes shared in all three species (left), in two species (middle), or in just one species (right). Data for *S. cerevisiae* from **Chu et al. (1998)**, data for *K. lactis* and *P. pastoris* generated in this study. A comparison of the 25 shared sporulation genes (**A**) and the seven shared Ndt80-dependent genes (**B**) shown in **Figure 6—figure supplement 1**.

The following figure supplements are available for figure 6:

**Figure supplement 1.** Comparing sporulation-induced genes and Ndt80-dependent genes across *S. cerevisiae*, *K. lactis*, and *P. pastoris*.

**Figure supplement 2.** Comparing patterns of meiotic expression for genes required for sporulation in *S. cerevisiae*.

missing from the genomes of species such as *C. lusitaniae* that are known to undergo meiosis and sporulation (**Bennett and Johnson, 2003**; **Sherwood and Bennett, 2009**). These observations, taken together, indicate that there is no 'universal' solution to meiosis and sporulation in terms of both the genes required for this process and the regulatory network itself. Instead, many different networks can apparently orchestrate this ancient process.

This work illustrates how a transcription regulator with hundreds of ancestral connections can, nonetheless, completely shift its cellular function. We propose that the continuous exploration of regulatory connections available to Ndt80—while still maintaining its role in sporulation—facilitated it reaching a network configuration that supported a new role in biofilm regulation. Computational models had previously led to this view, namely that gene regulatory networks are inherently flexible, allowing for the exploration of many configurations that maintain an overall function (**Barve and Wagner, 2013**; **Wagner, 2005a**). This exploration allows networks to sample many points in 'network space', some of which may be only a few changes away from generating a novel output. We believe that Ndt80 represents a tangible example of this idea: we propose that it is only through the inherent flexibility of the Ndt80 connections that the network—while still maintaining sporulation

and meiosis—could reach a position where the initial transition to regulating biofilm formation would be a simple one.

Although other transcription regulators may be subject to stronger evolutionary constraints than Ndt80, there is ample evidence that many transcription networks, like that of Ndt80, can drift through new configurations while preserving their transcriptional output (*Baker et al., 2012*; *Hare et al., 2008*; *Lavoie et al., 2010*; *Swanson et al., 2011*; *Tanay et al., 2005*; *Tsong et al., 2006*; *Villar et al., 2015*). Although the overall rate of network rewiring has not been extensively documented, it seems likely that the rewiring of the Ndt80 regulon is not unusually rapid when compared to at least some other fungal regulators (*Borneman et al., 2007*; *Tuch et al., 2008*). While much regulatory rewiring likely occurs through drift with no obvious shift in phenotype, we propose that continuous rewiring also predisposes regulators to a shift in function. Such extreme, inherent flexibility in gene regulatory networks, such as that exhibited by Ndt80, seems likely to be a general model to explain how complex regulatory networks can evolve to produce novel phenotypes.

## Materials and methods

### Media

All strains were grown in YPD at 30°C unless otherwise noted. For galactose induction, strains were grown in SRaffinose +1.7% galactose. For sporulation, *S. cerevisiae* strains were grown in liquid YPA (2% peptone, 2% potassium acetate, 1% yeast extract) and incubated at room temperature for 20–30 hr. For *K. lactis*, saturated liquid cultures in YPD were spotted onto SPO plates (1% potassium acetate, 2% agar + amino acids) and incubated at room temperature for 3 days. For *P. pastoris*, cells growing on YPD plates were patched onto 0.5% sodium acetate, 1% potassium chloride, 1% glucose, 2% agar plates and incubated at room temperature for 3 days. For *C. lusitaniae*, cells growing on YPD plates were patched onto PDA plates (0.37% potato dextrose +1.45% agar) and incubated at room temperature for 4 days. To quantify sporulation efficiency, three technical replicates (the same strain, grown up and plated on sporulation media independently) were performed and 200 cells were counted for each sample by DIC microscopy. In addition, a biological replicate (an independently-generated deletion strain) was assayed to confirm the phenotype of the first biological replicate. Nuclear staining using DAPI (*K. lactis*, *P. pastoris*) and Hoecsht (*C. lusitaniae*) was used to verify spore formation in each experiment.

### Strain construction

Strains used in this study are listed in *Supplementary file 2A* and primers used for strain construction are listed in *Supplementary file 2B*. Gene disruption cassettes for Ndt80 deletions were constructed by fusion PCR. For *K. lactis*, two cassettes were constructed, one with a *URA3* marker, one with a KanMX marker, each with 700 bp homology flanking the markers on either side. These constructs were transformed, sequentially, into a wild-type diploid strain. For *P. pastoris* and *C. lusitaniae*, a split marker approach was used and two constructs were generated for each gene disruption. For *P. pastoris*, a Hygromycin resistance marker and 700 bp flanking homology were used, and the split markers were transformed into two haploid strains of complementary mating types. These strains were then mated to form a diploid *ndt80* deletion (ploidy was verified by FACS). For *C. lusitaniae*, a NAT resistance marker was used along with 1 kb of flanking homology. The split markers were transformed into a wild-type α-cell, that was then mated with a wild-type **a**-cell, sporulated, and *ndt80* deletion **a**-cells were picked and verified by PCR. These were then mated with the *ndt80* deletion α-cell to generate a diploid *ndt80* deletion strain.

Tagged strains for ChIP were generated using a 13x C-terminal Myc tag (*Longtine et al., 1998*) inserted into the genome at the C terminus of the endogenous Ndt80 protein sequence. For *S. cerevisiae*, long primers were used to amplify the Myc tag fused to a KAN marker from pFA6a-13Myc-kanMX6 (*Longtine et al., 1998*), and this construct was integrated into a wild-type W303 strain. For *K. lactis*, split marker constructs were generated with 700 bp flanking homology fused to the Myc-KAN cassette and transformed into a wild-type diploid strain. For *P. pastoris*, split marker constructs were generated with 700 bp flanking homology fused to the Myc-NAT construct amplified from pADH34 (*Hernday et al., 2010*). In *S. stipitis*, homologous recombination was found to be very inefficient, and thus, constructs were randomly integrated into the genome. Constructs containing a

NAT marker upstream of an *E. gossypii* Tef1 promoter upstream of *S. stipitis* Ndt80 fused to a 13x C-terminal Myc tag with a Sat1 terminator were generated by PCR for both Ndt80A and Ndt80B and transformed into a wild-type strain. Two independent isolates were generated for each paralog and tested independently by ChIP-Seq to verify that the location of integration did not affect Ndt80 function or genomic binding. To test binding of *P. pastoris* Ndt80 in *S. cerevisiae*, fusion PCR was used to generate a construct with a Hygromycin resistance marker fused to the Gal1 promoter from *S. cerevisiae* upstream of *P. pastoris* Ndt80 fused to a 13x C-terminal Myc tag with a Sat1 termina- tor. This was fused to homology to *URA3* and integrated in an *S. cerevisiae* wild-type at the *URA3* locus. All Myc-tagged Ndt80 strains in S. cerevisiae, *K. lactis*, and *P. pastoris,* were tested for their ability to sporulate to ensure the tag did not interfere with endogenous Ndt80 function.

To generate strains with highly expressed Ndt80, different constitutive promoters were used, and constructs generated by fusion PCR to integrate these upstream of Ndt80. In *S. cerevisiae*, pGal1 from *S. cerevisiae* was used; in *K. lactis* pGal1 from *K. lactis* was used; in *P. pastoris* and *S. stipitis* pTef1 from *E. gossypii* was used; in *C. albicans* pTDH3 from *C. albicans* was used. In all cases, RT- qPCR was performed on the resulting strains as well as a wild-type strain to ensure that the constitu- tive promoters were indeed driving high Ndt80 expression.

*S. cerevisiae* and *C. albicans* were transformed using standard lithium acetate protocols. Electro- poration protocols were used for *K. lactis* (*Booth et al., 2010*), *P. pastoris* (*Cregg et al., 2009*), and *S. stipitis*. The protocol for *S. stipitis* was adapted from *Cregg et al. (2009)*, with cells harvested at an OD of 1.3–1.5.

## Genome sequences, gene annotations, and orthology mapping

Genome sequences and gene annotations for *S. mikatae*, *S. kudriavzevii*, *K. lactis*, *E. gossypii*, and *E. cymbalariae* were downloaded from the Yeast Gene Order Browser (YGOB) (*Byrne and Wolfe, 2005*). Genome sequences and gene annotations for *S. cerevisiae*, *C. lusitaniae*, *S. stipitis*, *C. dublin- iensis*, *C. tropicalis*, and *C. albicans* were downloaded from the Candida Gene Order Browser (CGOB) (*Maguire et al., 2013*). The genome sequence of *P. pastoris* strain CB7435 (*Küberl et al., 2011*) was downloaded from NCBI (accession number: PRJEA62483), and gene annotations were generated using the Yeast Genome Annotation Pipeline (*Proux-Wéra et al., 2012*). Gene annota- tions for YGOB, CGOB, and YGAP also include synteny-based orthology mapping, which was used to compare genes across species.

## Chromatin immunoprecipitation and high-throughput sequencing

For all ChIP experiments with endogenous or pTef1 promoters driving Ndt80 expression, samples were isolated from log-phase cultures grown in YPD and chromatin immunoprecipitation was per- formed as previously described (*Hernday et al., 2010*) using an anti-Myc monoclonal antibody (RRID: AB_2536303). Libraries were prepared using NEBNext Multiplex Kit for Illumina as previously described (*Sorrells et al., 2015*) and sequenced on an Illumina HiSeq 4000. For the mid-sporulation ChIP in *S. cerevisiae*, cells were induced to sporulate as previously described (*Chu and Herskowitz, 1998*; *Hepworth et al., 1998*), and samples were isolated after 16 hr in sporulation media. For ChIP of strains with pGal1 driving Ndt80 expression, strains were grown overnight in SRaffinose, diluted back and grown until log-phase in SRaffinose, then grown in SRaffinose +1.7% galactose for 5 hr.

For each ChIP-Seq experiment, a control experiment was performed using a matched strain miss- ing the C-terminal Myc tag. Two biological replicates (independently grown single-colonies of the same strain) were performed for both the tagged strain and the control untagged strain to identify regions bound by Ndt80 for that strain and condition.

## RNA sequencing in *P. pastoris* design and analysis

RNA expression was measured in *P. pastoris* using RNA-Seq. Samples were isolated from log-phase cultures in YPD or from mid-sporulation cultures after 30 hr. Total RNA was isolated using the Ambion RiboPure kit, and mRNA was isolated using the Oligotex mRNA Mini kit. Purified mRNA was then concentrated using the Zymo RNA Clean and Concentrator kit. Sequencing libraries were prepared using the NEBNext Ultra Directional RNA Library Prep Kit and sequenced on an Illumina HiSeq 4000. Three technical replicates (independently-grown single colonies of the same strain) were performed for each strain.

Reads were aligned to the genome using TopHat (*Trapnell et al., 2009*). Transcripts were then assembled, abundance was estimated, and differential expression was detected using Cufflinks (*Trapnell et al., 2013*).

## Full genome expression microarray in *K. lactis*

Custom-designed *K. lactis* microarrays were used to measure differential expression of genes in sporulation vs. mitotic growth and in an *ndt80* deletion vs. wild-type in sporulation media as previously described (*Booth et al., 2010*). Samples were isolated from log-phase cultures in YPD or from mid-sporulation cultures after 21 hr. Two technical replicates (independently-grown single colonies of the same strain) of each experiment were performed and the average of these replicates was taken.

## Comparing gene expression across *S. cerevisiae*, *K. lactis*, and *P. pastoris*

To identify genes upregulated in sporulation (*Figure 6—figure supplement 1*), different thresholds were used to compare gene expression in a wild-type strain grown in YPD to a wild-type strain grown in sporulation media. For *S. cerevisiae*, this threshold was a 2.5-fold expression increase in sporulation vs. mitotic growth, in a published dataset (*Chu et al., 1998*). For *K. lactis*, the threshold was a 1.5-fold expression increase in sporulation vs. mitotic growth in expression microarrays (described in detail above). For *P. pastoris*, the threshold was a statistically significant increase in sporulation vs. mitotic growth, as determined by analysis using Cuffdiff (*Trapnell et al., 2013*).

Similarly, to identify genes exhibiting Ndt80-specific expression in sporulation (*Figure 6—figure supplement 1*), expression was compared between a wild-type strain grown in sporulation media and an *ndt80* deletion strain grown in sporulation media. Different thresholds were applied to identify genes showing Ndt80-dependent expression: at least two-fold expression increase in WT vs. *ndt80* mutant for *S. cerevisiae*, at least 1.5-fold expression increase in in WT vs. *ndt80 mutant* for *K. lactis*, and a statistically significant increase (*Trapnell et al., 2013*) in *P. pastoris*.

## Biofilm experiments

Biofilms were grown and imaged by confocal scanning laser microscopy (CSLM) similar to previously described (*Mancera et al., 2015*). In brief, silicon squares were pre-incubated overnight in bovine serum at 30°C, washed with phosphate-buffered saline (PBS), and then submerged in Spider media containing 1% glucose instead of mannitol. Then cells from a culture grown overnight in YEPD at 30°C were added to the silicone squares upon reaching an $OD_{600}$ of 0.5. The squares were incubated for 90 min at 30°C shaking at 200 rpm for cell adherence. After adherence the squares were washed with PBS and transferred to fresh Spider 1% glucose and incubated for 48 hr at 30°C shaking at 200 rpm. For CSLM, the biofilms grown on the silicon squares were stained with concanavalin A Alexa Fluor 594 conjugate (50 μg/ml) and visualized using a Nikon Eclipse C1si upright spectral imaging confocal microscope and a 40x/0.80W Nikon objective.

## Identifying regions of Ndt80 binding

Sequencing reads were aligned to the genome using Bowtie (*Langmead et al., 2009*). Alignments were converted for visualization using SAMtools (*Li et al., 2009*). Peaks and fold enrichment values were generated using MACS2 (*Zhang et al., 2008*) with peak shift sizes generated using SPP (*Kharchenko et al., 2008*). Peaks were assigned to genes if any portion of the peak overlapped with the intergenic region upstream of that gene using MochiView (*Homann and Johnson, 2010*). For the heterologous *P. pastoris* Ndt80 in *S. cerevisiae* ChIP, fold enrichment values were mapped to genes by taking the maximum fold enrichment value for each intergenic region and mapping that to neighboring genes, also using MochiView.

Two biological replicates were performed for each ChIP-Seq experiment. Genes with Ndt80 peaks were identified for each replicate, and only genes with peaks in both replicates were considered as bona fide targets. For fold enrichment per gene calculations, the average of the maximum fold enrichment per intergenic region in each replicate was taken as the fold enrichment for neighboring genes.

Ndt80 targets were identified using several different criteria (*Figure 1C*). For Criteria 1, a gene was considered to be bound by Ndt80 if a peak was present in the intergenic upstream of that gene. For Criteria 2, a gene was considered to be bound by Ndt80 if a peak was present in upstream intergenic region and a consensus motif (CACAAA) was also present in that intergenic region (we did not require the motif to overlap the peak location). For Criteria 3, a gene was considered to be bound by Ndt80 if a peak was present in the upstream intergenic region, and a consensus motif was present in the intergenic of that species as well as intergenic regions upstream of orthologous genes in two closely related species. For *S. cerevisiae*, *S. mikatae* and *S. kudriavzevii* were used in this analysis; for *K. lactis*, *E. gossypii* and *E. cymbalariae* were used; and for *C. albicans*, *C. tropicalis* and *C. dubliniensis* were used (*Figure 2—figure supplement 1*). For Criteria 4, a gene was considered to be bound by Ndt80 if a peak was present in the upstream intergenic region, a consensus motif was also present in the upstream intergenic region, and the gene exhibited Ndt80-dependent expression. In *S. cerevisiae*, this was defined as at least a twofold change in expression in an *ndt80* deletion strain (up or down) compared with a wild-type strain during middle-meiosis (*Chu et al., 1998*). In *K. lactis*, this was defined as at least a 1.5-fold change in expression in the average of two replicates of an *ndt80* deletion compared to wild-type, as measured by expression array. In *P. pastoris*, this was defined as a statistically significant change in expression, as measured by Cufflinks (*Trapnell et al., 2013*) between three replicates of an *ndt80* deletion strain and three replicates of a wild-type strain. In *C. albicans*, this was defined as at least a 1.5-fold change in expression between an *ndt80* deletion and wild-type (*Nobile et al., 2012*).

## DNA *cis*-regulatory sequence motif discovery and enrichment

Ndt80 motifs were generated de novo for each ChIP-Seq experiment using DREME (*Bailey, 2011*). The union of peak locations in two biological replicates for each experiment were submitted to DREME, with a random set of genomic sequences of the same length as a negative control. The motif most closely matching the known Ndt80 binding site from *S. cerevisiae* (*Jolly et al., 2005*) and *C. albicans* (*Nobile et al., 2012*) was recorded along with the accompanying e-value (*Figure 2—figure supplement 4*).

A consensus Ndt80 motif across all species was generated using the union of replicate peak locations for all experiments performed with highly-expressed Ndt80. For *C. albicans* and *S. stipitis*, the intersection of replicate peaks was taken for each Ndt80 paralog, and then the union of paralog peak locations were taken to represent Ndt80 binding in that species. Peak locations for all five species tested were submitted to DREME, using a shuffled set of the same sequences as a negative control. The resulting motif (*Figure 2—figure supplement 4*) was trimmed by one base pair to give the high-confidence 'consensus motif' using for identifying high-confidence Ndt80 binding: CACAAA.

To determine enrichment of the consensus motif in binding locations for each ChIP-Seq experiment, compared to a random genomic background, a Fisher's one-tailed exact test was performed. The number of motifs in the peak locations for an experiment was compared to the number of motifs in a set of randomly generated genomic sequences of the same length to generate a p-value representing enrichment (*Figure 2—figure supplement 4*).

## Phylogenetic tree building

A phylogenetic tree of relevant species was constructed as previously described (*Lohse et al., 2013*). Protein sequences for 73 orthologs present in a single copy in all species were concatenated and aligned using MUSCLE (*Edgar, 2004*) and a tree was constructed using PHYML (*Guindon et al., 2010*). Two outgroup species, *N. crassa* and *A. nidulans* were used in the building of the tree to improve root placement, but were omitted from the tree image (*Figure 2A*) for simplicity.

A phylogenetic tree of Ndt80 protein sequences (*Figure 2—figure supplement 1*) was similarly constructed by aligning all Ndt80 protein sequences using MUSCLE (*Edgar, 2004*) and using PHYML to build a tree (*Guindon et al., 2010*). Similarity matrix of protein sequences (*Figure 2—figure supplement 1*) was generated by MUSCLE.

## Estimating sequence conservation between species

To determine overall sequence conservation between two species (*Figure 5B*, *Figure 5—figure supplement 1*), branch lengths from the phylogenetic species tree were used (*Figure 2A*). The additive

branch lengths separating two species from the most recent common ancestor were calculated for each two-species comparison, and then this was normalized to the value for *S. cerevisiae* and *K. lactis* (arbitrarily set to 1). This number was then multiplied by the fraction of Ndt80 targets conserved for any given two-species comparison to get a divergence time-corrected value.

## Simulation of overlap in Ndt80 ChIP-Seq targets across all species

In order to determine whether the 10 genes shown to be Ndt80 targets in all five species represented more than expected by chance (based on the number of targets in each species), we performed 100,000 simulations. In each simulation, genes were randomly selected from the pool of 3171 genes with orthologs in all five species, to represent the Ndt80 targets in all five species. For *S. cerevisiae*, 424 genes were selected, for *K. lactis* 269, for *P. pastoris* 300, for *S. stipitis* 958, and for *C. albicans* 989, as these are the number of mappable targets of Ndt80 in each species according to Criteria 2 (*Figure 1C*). For 100,000 simulations, the median number of genes found in all five sets was zero, and the maximum was five. Ten or more genes were not found in any round of the simulation (p-value<0.00001).

## Data deposition

ChIP-Seq and RNA-Seq data has been deposited to the NCBI Gene Expression Omnibus (GEO) repository under accession numbers GSE90660 (*S. cerevisiae* mitotic ChIP-Seq), GSE90661 (*S. cerevisiae* meiotic ChIP-Seq), GSE90662 (*K. lactis* ChIP-Seq), GSE90663 (*P. pastoris* ChIP-Seq), GSE90664 (*S. stipitis* ChIP-Seq), GSE90665 (*C. albicans* ChIP-Seq), and GSE92766 (RNA-Seq).

## Acknowledgements

We thank C Dalal, S Coyle, C Baker, C Britton, S Singh-Babak, and L Pack for valuable comments on the manuscript; T Sorrells and C Nobile for important technical contributions and comments on the manuscript; E Chow and the UCSF CAT for expert advice; and R Bennett and R Sherwood for strains and protocols. This work was supported by grant R01 GM037049 from the National Institutes of Health. EM was supported by the Human Frontier Science Program and UC-MEXUS.

## Additional information

### Funding

| Funder | Grant reference number | Author |
| --- | --- | --- |
| National Institutes of Health | R01 GM037049 | Isabel Nocedal<br>Eugenio Mancera<br>Alexander D Johnson |
| Human Frontier Science Program | | Eugenio Mancera |
| UC-MEXUS | | Eugenio Mancera |

The funders had no role in study design, data collection and interpretation, or the decision to submit the work for publication.

### Author contributions

IN, Conceptualization, Formal analysis, Investigation, Writing—original draft, Writing—review and editing; EM, Formal analysis, Investigation; ADJ, Conceptualization, Formal analysis, Supervision, Writing—review and editing

### Author ORCIDs

Isabel Nocedal, http://orcid.org/0000-0002-4706-1113

## Additional files

### Supplementary files
• Supplementary file 1. Excel spreadsheet containing processed data for all ChIP-Seq, RNA-Seq, and expression array experiments performed. Rows correspond to groups of orthologous genes. Columns show results of all genomic experiments discussed here, colors correspond to experiments in different species. Cell value of 'N/A' indicates no orthologous gene for that orthogroup in that species.

• Supplementary file 2. Excel spreadsheet containing strains and primers used in this study. (A) Strains used in this study. (B) Primers used in this study.

### Major datasets
The following previously published dataset was used:

| Author(s) | Year | Dataset title | Dataset URL | Database, license, and accessibility information |
|---|---|---|---|---|
| Chu S, DeRisi J, Eisen M, Mulholland J, Botstein D, Brown PO, Herskowitz I | 1998 | The transcriptional program of sporulation in budding yeast | https://www.ncbi.nlm.nih.gov/sites/GDSbrowser?acc=GDS104 | Also available at the NCBI Gene Expression Omnibus (accession no: GDS104) |

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
