## [Decision Letter]

Thank you for submitting your article "Extreme gene regulatory network plasticity facilitates a switch in function of a conserved transcription regulator" for consideration by *eLife*. Your article has been reviewed by three peer reviewers, and the evaluation has been overseen by Patricia Wittkopp as both Senior and Reviewing Editor. The following individuals involved in review of your submission have agreed to reveal their identity: Scott Barolo (Reviewer #1) and Zhenglong Gu (Reviewer #3).

The reviewers have discussed the reviews with one another and the Reviewing Editor has drafted this decision to help you prepare a revised submission.

Summary:

All three reviewers found this work of potential interest for publication in *eLife*, but also identified a number of issues that must be addressed prior to publication. These issues, which all reviewers and the reviewing editor agreed to in our post-review discussion, are summarized below, with some of the reviewers originally submitted comments worked in for context.

Essential revisions:

1) The conclusion that evolutionary plasticity of the Ndt80 network "facilitated" the change in Ndt80 function is not well supported and must be toned down. In the words of one reviewer: "At most, it seems to me the authors could say that the acquisition of the new function is consistent with their model of plasticity facilitating the evolution. All they've shown is that the two things are true - I don't see anything that connects the first to the second. To actually provide evidence for this claim, the authors would have to show, e.g., that at least that the plasticity in Ndt80 is unusual, as is the divergence of function. Alternatively, maybe they could show some kind of theory or simulation showing that this type of evolution would be less likely if the evolution of the regulatory network were less plastic. However, without getting deeply into the molecular/cellular mechanisms of the biofilm formation in *C. albicans*, I don't see how the authors could really provide direct support for this claim. I don't think this additional work is within the scope of this paper. The authors could also reduce the claim that the rapid divergence "facilitated" the evolution of the phenotype. Another way to see the observations of this paper is that the divergence in the transcriptional network appears relatively constant, even though the phenotypic change is lineage specific. Perhaps this is the surprising finding: transcriptional rewiring doesn't seem to correlate with phenotypic change."

2) The paper should do more to put this work in the context of other work on evolution of regulatory networks. Particularly, the work on the evolution of the regulation of the yeast ribosome needs to be discussed. This work showed how conserved regulators could be co-opted into new regulatory functions. Also, related to this "Yet it remains unclear how large numbers of genes (whether old or new) become incorporated into an emerging regulatory network. It is also not known how an ancient, conserved transcription regulator can form the basis of an entirely new regulatory network." I see the point the authors are trying to make here, but there have been some studies that give at least some hints as to how these things happen. For example, previous work from the Johnson lab in yeast and David Stern and Sean Carroll's labs in flies showed how new targets get incorporated into existing networks. We also know from work on the yeast ribosome that highly conserved processes (like translation) can change their regulatory mechanisms over evolution. The authors should put their work in this context:

Tanay A, et al. Proc Natl Acad Sci U S A. 2005 May 17;102(20):7203-8

Ihmels et al. Science. 2005 Aug 5;309(5736):938-40

Hogues Het al. Mol Cell. 2008 Mar 14;29(5):552-62.

3) Additional support is needed to claim that Ndt80s rewiring is "rapid" or "extensive". Specifically, how does this rewiring compare to that of other genes? "Because several recent multi-species ChIP studies have showed evidence for divergence of binding, and because the amount of divergence inferred depends on the analysis methods used, I'm not sure how the authors observations about Ndt80 compares to what would be expected for a "normal" transcription factor (that didn't undergo rapid or extensive divergence)." Also, how much "divergence" is expected from biological replicates (in the same species) and how this compares to what is observed between species? This will reveal whether 13% or 2% (from different analysis methods) is "little" overlap. This puts a lower bound on how much "evolution" is expected due to experimental noise alone. (e.g., PMID:26429922, Figure 3). We suggest addressing this by comparing your estimates for Ndt80 to data from other transcription factors using the same data analysis pipeline. This could be done using previously published data.

4) Correct (or clarify) interpretation of paralogs: "While the NDT80 genes in *S. cerevisiae* and *C. albicans* discussed above are orthologous, *C. albicans* has an additional paralog of Ndt80, resulting from a recent gene duplication (Figure 2, (Sellam et al. 2010)). The Ndt80 ortholog whose targets we identified above will be referred to as Ndt80B, while the additional paralog will be referred to as Ndt80A (Figure 2—figure supplement 1)." The tree in Figure 2—figure supplement 1 does not support this. Ndt80B appears to be the derived form, while Ndt80 appears ancestral. The authors should clarify the evidence that they have the correct homolog (or if there is a "correct" one). Further, the duplication does not appear recent, as claimed in the text. Most of the species in the clade containing *C. albicans* contain the duplicate. In the main text, for support of the "recent" duplication, it points to Figure 2. In the figure legend is says: "Phylogenetic tree of the species investigated, inferred from protein sequences of 73 highly conserved genes (Lohse et al. 2013) with the scale representing the number of nucleotide substitutions per site." This must be wrong because the nucleotide divergence between *S. cerevisiae* and *K. lactis* is already very large (>2 subs. per site). It's not really possible to measure nucleotide divergence between such distantly related species as shown in this tree.

"In *C. albicans* and *S. stipitis*, there are two paralogs of Ndt80 and so targets were identified for each paralog separately and the union was taken to represent all Ndt80 targets in that species." I don't understand how this is consistent with "While the NDT80 genes in *S. cerevisiae* and *C. albicans* discussed above are orthologous". At the very least the authors need rethink and clarify their treatment of the paralogs.

5) "This observation strongly supports the conclusion that the majority of the differences in the Ndt80 regulons across species are due to gains and losses of the conserved Ndt80 cis-regulatory sequence." I think this is a great experiment, but I think this claim is overstated. The experiment shows that the differences in regulation are not due to changes in the protein sequence of Ndt80. It doesn't rule out the possibility of additional trans-factors (e.g., different co-factors) in the different species that are responsible for the differences in Ndt80 target specificity. (In passing, I get what the authors are trying to say here, but I think it's very confusing to say gains and losses of a conserved sequence. How about gains and losses of consensus motif matches or cis-regulatory DNA binding sites?)

6) To explain "how Ndt80 could have maintained a role in sporulation despite the near complete turnover of its target genes", a hypothesis is proposed: "the genes required for sporulation and meiosis are themselves changing across species" (subsection “The transcription network that controls meiosis and sporulation changes rapidly”). This is a very interesting yet plausible hypothesis. To test it, the authors measured gene expression changes during sporulation in *K. lactis, P. pastoris*, and *S. cerevisiae* and found very little overlap (Figure 6). This is interesting and to the point, but it doesn't fully answer the question or test the hypothesis. Perhaps the 25 shared targets are the key ones for sporulation in all three species. If so, the authors' conclusion that there is no shared sporulation network is wrong. In the absence of functional data establishing which genes are necessary for sporulation in each species, the authors should back off from their firm conclusion that the key sporulation genes are different in each species.

---

## [Author Response]

*Essential revisions:*

*1) The conclusion that evolutionary plasticity of the Ndt80 network "facilitated" the change in Ndt80 function is not well supported and must be toned down. In the words of one reviewer: "At most, it seems to me the authors could say that the acquisition of the new function is consistent with their model of plasticity facilitating the evolution. All they've shown is that the two things are true - I don't see anything that connects the first to the second. To actually provide evidence for this claim, the authors would have to show, e.g., that at least that the plasticity in Ndt80 is unusual, as is the divergence of function. Alternatively, maybe they could show some kind of theory or simulation showing that this type of evolution would be less likely if the evolution of the regulatory network were less plastic. However, without getting deeply into the molecular/cellular mechanisms of the biofilm formation in C. albicans, I don't see how the authors could really provide direct support for this claim. I don't think this additional work is within the scope of this paper. The authors could also reduce the claim that the rapid divergence "facilitated" the evolution of the phenotype. Another way to see the observations of this paper is that the divergence in the transcriptional network appears relatively constant, even though the phenotypic change is lineage specific. Perhaps this is the surprising finding: transcriptional rewiring doesn't seem to correlate with phenotypic change."*

We agree with the reviewers that it is difficult to rigorously prove that the plasticity of the Ndt80 network facilitated the change in Ndt80 function. We have made several modifications to clarify that this idea is a proposal, rather than a conclusion. Most notably, we have changed the title of the paper from “Extreme gene regulatory network plasticity facilitates a switch in function of a conserved transcription regulator” to “Gene regulatory network plasticity predates a switch in function of a conserved transcription regulator. In addition, we have made changes to the Abstract (last sentence), Introduction (last two sentences), and Discussion (first sentence) to clarify that this idea is a proposal rather than a conclusion. In passing, it is worth noting that the counter hypothesis – that the plasticity did not play a role – seems unlikely to us. Nevertheless, we now properly distinguish between our conclusions and our model.

*2) The paper should do more to put this work in the context of other work on evolution of regulatory networks. Particularly, the work on the evolution of the regulation of the yeast ribosome needs to be discussed. This work showed how conserved regulators could be co-opted into new regulatory functions. Also, related to this "Yet it remains unclear how large numbers of genes (whether old or new) become incorporated into an emerging regulatory network. It is also not known how an ancient, conserved transcription regulator can form the basis of an entirely new regulatory network." I see the point the authors are trying to make here, but there have been some studies that give at least some hints as to how these things happen. For example, previous work from the Johnson lab in yeast and David Stern and Sean Carroll's labs in flies showed how new targets get incorporated into existing networks. We also know from work on the yeast ribosome that highly conserved processes (like translation) can change their regulatory mechanisms over evolution. The authors should put their work in this context:*

*Tanay A, et al. Proc Natl Acad Sci U S A. 2005 May 17;102(20):7203-8*

*Ihmels et al. Science. 2005 Aug 5;309(5736):938-40*

*Hogues Het al. Mol Cell. 2008 Mar 14;29(5):552-62.*

We agree with the reviewers that network rewiring is not a new idea, and we certainly did not mean to imply this. We have added a paragraph to the end of the Introduction (with numerous references) to better put our work in a broader context of other work on regulatory rewiring. We have also changed the two sentences specifically referenced by the reviewers in the Introduction (end of the first paragraph) to clarify the question that we believe this study addresses.

We do wish to point out that this paper differs from previous work in studying how a conserved regulator was co-opted to form the basis of a large, new circuit (comprising hundreds of genes), which produces a novel phenotype. This is in contrast to previous work that has focused primarily on changes in the regulation of conserved set of genes, such as the ribosomal or **a**-specific genes, in yeast.

*3) Additional support is needed to claim that Ndt80s rewiring is "rapid" or "extensive". Specifically, how does this rewiring compare to that of other genes? "Because several recent multi-species ChIP studies have showed evidence for divergence of binding, and because the amount of divergence inferred depends on the analysis methods used, I'm not sure how the authors observations about Ndt80 compares to what would be expected for a "normal" transcription factor (that didn't undergo rapid or extensive divergence)." Also, how much "divergence" is expected from biological replicates (in the same species) and how this compares to what is observed between species? This will reveal whether 13% or 2% (from different analysis methods) is "little" overlap. This puts a lower bound on how much "evolution" is expected due to experimental noise alone. (e.g., PMID:26429922, Figure 3). We suggest addressing this by comparing your estimates for Ndt80 to data from other transcription factors using the same data analysis pipeline. This could be done using previously published data.*

For many reasons, it is difficult to compare the rates of Ndt80 rewiring with previous work. For example, we have used numerous “stringency” criteria (ChIP, transcript profiling, and conservation of binding sites) to try to eliminate false positives, whereas other work (including our own) typically lacks these additional analyses. Our best guess is that the high rate of Ndt80 rewiring is not unusual; it is clearly much higher than that exhibited by Matα1 or the Mat**a**1-α2 heterodimer, but is in the ballpark range of that of Mcm1 (Tuch et al. 2008) and Gal4 (Askew et al. 2009). We now explicitly state this in the Discussion (last paragraph).

We believe that the rewiring of the Ndt80 regulon is notable not in its comparison to that of other regulators, but in comparison to its conservation in phenotype. In our experience, many scientists are still surprised that a conserved regulator that controls a conserved process exhibits any rewiring whatsoever. For this reason, we believe the use of the term “extensive” is appropriate, given the small overlap in genes controlled by Ndt80 across species. However, we can see how use of the term “rapid” may have implied a comparison with other regulators, and have chosen to substitute the term “rapid” with “extensive” when referring to the rewiring of the Ndt80 regulon (Abstract; subsection “The transcription network that controls meiosis and sporulation changes extensively” heading and first paragraph).

We appreciate the observation that we have neglected to address the issue of experimental noise directly in the manuscript, and agree that it is important to clarify that the differences that we observe between species are not attributable to simple noise in our ChIP-seq experiments. We have added a figure supplement (Figure 1—figure supplement 2), inspired by the suggested paper that shows the fraction of targets shared between biological ChIP-seq replicates of *S. cerevisiae* and *C. albicans*, and how these compare to the differences we observe between the two species, for all four methods of Ndt80 target identification. These results show that experimental noise alone results in less than a 20% difference in Ndt80 targets, and less than a 5% difference for our most stringent method of target identification. In contrast, the comparisons between *S. cerevisiae* and *C. albicans* show greater than 85% difference in the least stringent method of identification and greater than 95% for the most stringent, suggesting that experimental noise is not sufficient to explain the differences we observe. We have added a sentence explaining this and referring to the new figure supplement in the first Results section (subsection “Ndt80 target genes differ markedly between *S. cerevisiae* and *C. albicans*”, end of third paragraph).

*4) Correct (or clarify) interpretation of paralogs: "While the NDT80 genes in S. cerevisiae and C. albicans discussed above are orthologous, C. albicans has an additional paralog of Ndt80, resulting from a recent gene duplication (Figure 2, (Sellam et al. 2010)). The Ndt80 ortholog whose targets we identified above will be referred to as Ndt80B, while the additional paralog will be referred to as Ndt80A (Figure 2—figure supplement 1)." The tree in Figure 2—figure supplement 1 does not support this. Ndt80B appears to be the derived form, while Ndt80 appears ancestral. The authors should clarify the evidence that they have the correct homolog (or if there is a "correct" one). Further, the duplication does not appear recent, as claimed in the text. Most of the species in the clade containing C. albicans contain the duplicate. In the main text, for support of the "recent" duplication, it points to Figure 2. In the figure legend is says: "Phylogenetic tree of the species investigated, inferred from protein sequences of 73 highly conserved genes (Lohse et al. 2013) with the scale representing the number of nucleotide substitutions per site." This must be wrong because the nucleotide divergence between S. cerevisiae and K. lactis is already very large (>2 subs. per site). It's not really possible to measure nucleotide divergence between such distantly related species as shown in this tree.*

*"In C. albicans and S. stipitis, there are two paralogs of Ndt80 and so targets were identified for each paralog separately and the union was taken to represent all Ndt80 targets in that species." I don't understand how this is consistent with "While the NDT80 genes in S. cerevisiae and C. albicans discussed above are orthologous". At the very least the authors need rethink and clarify their treatment of the paralogs.*

We thank the reviewers for this comment, and have rewritten the section discussing the Ndt80 paralogs extensively to attempt to clarify our results related to the two Ndt80 paralogs. To clarify, it is not possible to definitively conclude which of the Ndt80 paralogs is more related to the ancestral pre-duplication Ndt80. While the reviewers are correct that the phylogenetic tree in Figure 2—figure supplement 1 indicates that Ndt80B is the original form, synteny-based analysis suggests that Ndt80A may in fact be the ancestral paralog. In addition, protein trees built using different alignment methods show different topologies, with some indicating Ndt80B is the ancestral form and others indicating that Ndt80A is the ancestral form. Because of this uncertainty, we have made no attempt to conclude which paralog descends from the original Ndt80 and which arose as a result of the gene duplication.

Instead, we have focused our analysis primarily on Ndt80B for two reasons: (1) Ndt80B is the gene required for biofilm formation in *C. albicans*, while Ndt80A has no known phenotype, and (2) the targets of Ndt80A are also targets of Ndt80B. We identified the genes bound by the two Ndt80 paralogs in both *C. albicans* and *P. stipitis* and found that, in both species, Ndt80A binds primarily to a small subset of the Ndt80B targets (Figure 1—figure supplement 4 and [Supplementary-material SD1-data]). This suggests that the change in Ndt80 targets that occurred since the divergence of *S. cerevisiae* and *C. albicans* was not a direct result of the gene duplication, and more likely (as we show later) occurred prior to the duplication. Because the respective roles of the two paralogs are unknown in *C. albicans* and *P. stipitis*, we have treated them, in a sense, as a single protein. Because Ndt80A binds primarily to a small subset of the Ndt80B targets, in both species we took the union of the paralog targets (encompassing all the Ndt80B targets and the few unique Ndt80A targets) as representative of the overall Ndt80 targets in that species. In this way, we feel we are remaining agnostic to the question of which is the “correct” paralog and are instead focusing on the species-specific divergence in overall Ndt80 targets. We note that this uncertainty does not affect any of our major conclusions.

However, we can see how this was not clear from the manuscript, and have made a number of changes to attempt to clarify this issue. We have added a new section in Results entitled “Divergence in Ndt80 targets not a result of Ndt80 gene duplication”, which includes text from the original manuscript with several changes. Notably, the reviewers were correct to point out that the use of the term “ortholog” and “orthologous” is incorrect in the discussion of a paralog, and this has been corrected to “homolog” and “homologous”. In addition, we have changed “recent gene duplication” to simply “gene duplication” as the term “recent” may serve to confuse more than clarify. We have also added several sentences to this section to clarify our conclusions about the paralogs. In addition, we have added a sentence later in the Results section to clarify that the paralogs in *P. stipitis* also bind highly overlapping targets, which is why we have considered the union of their targets in our cross-species comparisons (see subsection “Ndt80 target genes also differ between *S. cerevisiae, K. lactis, P. pastoris, S. stipitis*, and *C. albicans*”, first paragraph).

We also thank the reviewers for noticing an error in the caption of Figure 2 (also present in the captions for Figure 5 and Figure 5—figure supplement 1): “nucleotide substitutions per site” has been corrected to “substitutions per site”, as this rate refers to the rate of amino acid substitutions in the proteins used to construct the tree, not nucleotide substitutions.

*5) "This observation strongly supports the conclusion that the majority of the differences in the Ndt80 regulons across species are due to gains and losses of the conserved Ndt80 cis-regulatory sequence." I think this is a great experiment, but I think this claim is overstated. The experiment shows that the differences in regulation are not due to changes in the protein sequence of Ndt80. It doesn't rule out the possibility of additional trans-factors (e.g., different co-factors) in the different species that are responsible for the differences in Ndt80 target specificity. (In passing, I get what the authors are trying to say here, but I think it's very confusing to say gains and losses of a conserved sequence. How about gains and losses of consensus motif matches or cis-regulatory DNA binding sites?)*

We thank the reviewer for pointing this out. The result we obtained shows that the difference in Ndt80 targets is largely due to changes in cis-regulatory sequences, but we agree that we did not show that these changes occurred in the Ndt80 cis-regulatory sequence as opposed to the cis-regulatory sequence of a hypothetical co-factor. We have added a sentence to the Results section explaining this (subsection “Ndt80 binding differences are largely determined by the gain and loss of *cis*-regulatory sites”, last paragraph). We have also reworded the awkward construction noted.

*6) To explain "how Ndt80 could have maintained a role in sporulation despite the near complete turnover of its target genes", a hypothesis is proposed: "the genes required for sporulation and meiosis are themselves changing across species" (subsection “The transcription network that controls meiosis and sporulation changes rapidly”). This is a very interesting yet plausible hypothesis. To test it, the authors measured gene expression changes during sporulation in K. lactis, P. pastoris, and S. cerevisiae and found very little overlap (Figure 6). This is interesting and to the point, but it doesn't fully answer the question or test the hypothesis. Perhaps the 25 shared targets are the key ones for sporulation in all three species. If so, the authors' conclusion that there is no shared sporulation network is wrong. In the absence of functional data establishing which genes are necessary for sporulation in each species, the authors should back off from their firm conclusion that the key sporulation genes are different in each species.*

We appreciate the reviewer’s note that we have omitted any discussion of the genes required for sporulation in our discussion of our expression results. We have added a figure supplement (Figure 6—figure supplement 1), along with a sentence referring to this data in the Results section (subsection “The transcription network that controls meiosis and sporulation changes extensively”, last paragraph) that we believe supports our conclusion that the genes required for sporulation are changing across species. While we have not directly tested for the genes required for sporulation in *K. lactis* and *P. pastoris* (as this would be extremely labor intensive), we do find that 21 of the genes required for sporulation in *S. cerevisiae* that exhibit upregulation in sporulation are missing entirely from the genomes of *K. lactis* and *P. pastoris*. In addition, of those that are present in these two species (46), only 9 show sporulation up-regulation in both species tested. Based on these results, as well as the expression results shown in Figure 6, we believe there is strong evidence that there are significant changes in the genes involved in sporulation between these species. We do not believe, however, that there is “no shared sporulation network”, and have removed the last sentence from the Results section, in order to ensure that the reader does not draw this incorrect conclusion from our results.